# Damage Characteristics of PELE Projectile with Gradient Density Inner Core Material

**DOI:** 10.3390/ma11122389

**Published:** 2018-11-27

**Authors:** Liangliang Ding, Jingyuan Zhou, Wenhui Tang, Xianwen Ran, Ye Cheng

**Affiliations:** 1College of Liberal Arts and Sciences, National University of Defense Technology, Changsha 410073, China; dingliangliang14@nudt.edu.cn (L.D.); zhoujingyuan12@163.com (J.Z.); ranxianwen@nudt.edu.cn (X.R.); 2School of Basic Sciences for Aviation, Naval Aviation University, Yantai 264001, China; chengye2014@163.com

**Keywords:** PELE, penetration with enhanced lateral efficiency, gradient density, inner core, penetration ability, fragmentation effect

## Abstract

The PELE (penetration with enhanced lateral efficiency) projectile is a new type of safe ammunition which can form a large number of fragments after perforating the target, and does not depend on any pyrotechnics. The damage characteristics of PELE projectile mainly include the penetration ability and fragmentation effect. There are many factors affecting the damage characteristics of PELE projectile, and this paper attempts to study the damage characteristics of PELE projectile, from the point of view of changing the single core material. Therefore, four different inner core combination types were designed in this paper, namely, zero gradient—I type (PE), zero gradient—II type (Al), positive gradient type (PE + Al), and negative gradient type (Al + PE). With the help of a more mature numerical simulation method, the studies were carried out from several aspects, such as the axial residual velocity of projectile, the radial scattering velocity of fragments, the radial scattering radius of fragments, and the residual length of projectile. The axial residual velocity of projectile can characterize the penetration ability of projectile, the radial scattering velocity and radial scattering radius of fragments can predict the damage area of fragments, and the residual length of projectile can reflect the fragment conversion rate of casing. The results indicate that the negative gradient inner core combination is superior to the other three combinations in terms of the penetration ability and fragmentation effect. Under the same impact velocity, the maximum radial velocity and radial scattering radius of fragments mainly depend on the front inner core material, and these two parameters will increase appropriately with the increase of the strength of front inner core material. Similarly, the residual length of projectile can be reduced, or the fragment conversion rate can be enhanced, by properly reducing the strength of rear inner core material. The conclusions obtained in this paper can provide a reference for engineering applications.

## 1. Introduction

The PELE (penetration with enhanced lateral efficiency) projectile is a new type of safe ammunition proposed and developed in recent years [1,2]. This new type ammunition does not need fuze and explosive charge, but can realize the double terminal effect of armor piercing and fragment lethality only by the physical interaction between the projectile and target. A PELE projectile is mainly composed of outer casing and inner core in structure. When a PELE projectile impacts the target plate at relatively high velocity, the high-density casing penetrates the target plate first, and then the low-density inner core material is squeezed between the projectile and target plate, which results in high dynamic pressure inside the projectile. When the dynamic pressure reaches or exceeds the strength limit of the casing material, the casing will be cracked and break into a large number of fragments, due to stress unloading after perforating the target plate, which significantly enhances the damage power to the target. From the working principle of PELE projectile, it can be seen that in order to ensure the projectile has good penetration performance, the outer casing is usually composed of high-density metal materials, such as tungsten or steel; in order to ensure the inner core has good compressibility, the inner core material is usually composed of low-density materials, such as aluminum or plastic. The interaction between the projectile and target plate is shown in Figure 1. In short, the PELE projectile has the characteristics of converting axial kinetic energy into radial kinetic energy, and does not depend on any pyrotechnics, and it also has the lateral fragmentation effect while retaining the penetration ability of the traditional armor-piercing projectile [3,4,5].

As for the terminal effect of PELE projectile, many scholars have done a lot of work on the theoretical mechanism of PELE projectile. Paulus et al. [5] established a theoretical model to calculate the axial residual velocity and radial scattering velocity of fragments after perforating the target, and a series of tests were designed to verify the model. Zhu et al. [6] and Du et al. [7] analyzed the fragmentation effect of projectile after perforating the target by means of experiments and theory. According to the Mott fragment distribution theory [8,9] and the shock wave theory, Verreault et al. [10,11,12] and Fan et al. [13] established the theoretical model to estimate the radial scattering velocity of PELE projectile after perforating the target plate separately. In addition, the effects of the casing and core materials, the interaction conditions between projectile and target, and other factors on the penetration behavior and fragmentation effect of PELE projectile, have also been carried out in many of research studies. Zhu et al. [14,15] and Ye et al. [16] studied the effect of impact velocity of projectile, the Poisson’s ratio, and elastic modulus of inner core material, the density, and compressive-tensile strength of outer casing on the fragmentation effect of PELE projectile. Tu et al. [17,18] studied the effect of target thickness, impact velocity, and casing/core material combination on the penetration ability of PELE projectile. Jiang et al. [19] and Yin et al. [20] studied the terminal effect of PELE projectiles with different inner core materials by experiments and numerical simulations, respectively.

According to the published literature, almost all of these studies were carried out for single inner core material, and there are few published papers on the PELE penetration behavior of multi-core material [21]. In fact, during the process of PELE penetrating the target plate, the high dynamic pressure distribution generated inside the projectile is extremely uneven, and the dynamic pressure of the projectile head is significantly higher than that of the projectile rear, which will greatly affect the lateral effect of fragments. Therefore, this paper expects to study and analyze the mechanical behavior and lateral enhancement effect of the PELE projectile with gradient density inner core material by means of AUTODYN-3D numerical simulation. Finally, it is hoped that an optimum scheme of inner core combination can be found to provide guidance and reference for the structural design of PELE projectile in engineering application.

## 2. Analysis of the Fracture Process of PELE Projectile

When the PELE projectile impacts the target plate, the axial impact force will be generated on the inner core, due to the obstruction of target plate, as shown in Figure 2a. Under the impact, the shock wave is formed in the inner core, so that the inner core is axially compressed, as shown in Figure 2b. The axially compressed inner core undergoes radial expansion under the influence of Poisson effect, thereby subjecting the outer casing to circumferential stretching and radial shearing. If the radial shearing action of outer casing is neglected, the outer casing can be considered to be formed by a plurality of rings stacked axially along the projectile. At this point, the stress state of the casing can be approximately equivalent to the superposition of many rings, with inner wall subjected to radial force, as shown in Figure 2c. According to Mott’s dynamic fracture theory, when the circumferential tensile strain reaches the fracture ultimate strain of the outer casing material, the casing material first cracks on the outer surface. After the crack occurs, the unloading wave is generated at the crack and propagates to both sides, so that the tensile stress in the casing material is unloaded. The strength of the unloading wave increases with the development of the crack into the interior of casing material. When the crack perforates the inner surface, the unloading wave intensity reaches the maximum. If the casing is an ideal rigid plastic material, the unloading stress on the wave front, at this moment, reaches the yield stress of material. With the expansion of outer casing, the crack continues to extend into the interior of casing until it completely perforates the casing to form a fracture surface. When the crack of outer casing is completed, the unloading area formed by the two adjacent fracture surfaces forms a fragment, as shown in Figure 2d.

Due to the influence of the axially transmitted shock wave, the circumferential crack generated in the front-end casing will continue to develop along the axial direction toward the rear-end of the projectile, so that the outer casing is formed into a plurality of independent elongate fragment strips. Under the joint action of the radial expansion force of the inner core and outer casing, the moving speed of radial expansion is sequentially obtained, from the front to the rear, in the axial direction. Therefore, the two adjacent axial particles in the fragment zone near the shock front have different radial speeds, due to the different time starting point of radial acceleration, so that the two adjacent axial particles produce shear slip in the process of radial expansion. When the shear slip reaches a certain extent, the fragment zone will fracture axially, which makes the fragment zone form fragments with a certain length.

In order to facilitate the description of each part of PELE projectile and fragments, the following definitions are given below, as shown in Figure 3. The contact end of projectile is called the projectile head, and the other end is called the projectile rear. The crack formed along the circumferential plane of the casing is called the circumferential crack, and the fracture in this direction is called the circumferential fracture. The crack formed along the axial direction of the casing is called the axial crack, and the crack in this direction is called the axial fracture. The fragment size between the two circumferential fracture surfaces is called the fragment width, and the fragment size from the projectile head to the axial crack, or between the two axial cracks, is called the fragment length, and the radial thickness of the casing is called the fragment thickness.

## 3. Numerical Simulation

A large number of studies have shown that the density, Young’s modulus, and Poisson’s ratio of the inner core material have a certain influence on the PELE projectile penetration behavior and the fragmentation effect. It is precisely because the difference in the properties of the inner core material has a great influence on the fragmentation effect of PELE projectile, and this paper attempts to study whether the damage effect of PELE projectile can be optimized by designing different inner core material combinations types. In order to facilitate the preliminary research and analysis, this paper only divides the inner core material into two parts, and realizes the inner core material combinations types by changing the density of the two inner core materials. The wave impedance of the inner core can reflect the ability to resist deformation after impacting the target plate. The greater the wave impedance, the weaker the ability to laterally expand. The magnitude of the wave impedance can be expressed as the product of the material density and the volumetric sound velocity. The inner core density near the impact end is defined as *ρ*_1_, and the volumetric sound velocity is defined as *c*_1_; the inner core density near the projectile rear is defined as *ρ*_2_, and the volumetric sound velocity is defined as *c*_2_. When the two inner cores are of the same material, that is, there is no gradient, then two types of zero gradient are defined. When the density (wave impedance) of homogeneous inner core material is low, it is defined as zero gradient—I type, as shown in Figure 4a. When the density (wave impedance) of homogeneous inner core material is high, it is defined as zero gradient—II type, as shown in Figure 4b. When *ρ*_1_*c*_1_ < *ρ*_2_*c*_2_, the inner core combination type is defined as positive gradient, as shown in Figure 4c. When *ρ*_1_*c*_1_ > *ρ*_2_*c*_2_, the inner core combination type is defined as negative gradient, as shown in Figure 4d.

### 3.1. Finite Element Model

In order to facilitate analysis, the entire simulation model is simplified into four parts: outer casing, front inner core, rear inner core, and target plate. The length of the PELE projectile is *l*_0_ = 50 mm, and the diameter is *D* = 10 mm. The diameter of the inner core is *d* = 6 mm, and the length of the front and rear inner core depends on the specific working conditions. The length and width of the target plate are *ψ* = *L* = 120 mm, and the thickness of the target plate is *h* = 3 mm. The geometric schematic diagram and the finite element model are shown in Figure 5 and Figure 6.

In this paper, the mesh generation of the whole model was completed by the HyperMesh 13.0 software (Altair Engineering, Detroit, MI, USA), and the solutions were solved by the nonlinear dynamics software AUTODYN v14.5, which was developed by ANSYS Inc. (Canonsburg, PA, USA). The mesh type is unstructured hexahedral mesh, and the average mesh size is 0.25 mm. In order to improve the computational efficiency, the meshes of the target plate adopted the variable-step size to reduce the number of meshes. The main idea of gradient mesh is that the central area is denser, the boundary area is sparse, and the intermediate mesh is gradient, as shown in Figure 6d. Considering the symmetry characteristics of the model under vertical penetration, the model adopted a 1/4 simplification. In order to facilitate the analysis of the interaction between the projectile and target plate, a series of Gaussian points were arranged along the axial direction on the outer surface of the casing, and the distance between the points is 2 mm. The interaction algorithm of the whole model adopted the Lagrange/Lagrange algorithm, and the transmissive boundary condition was applied to the edge of the target plate. The literature [22] introduced the stochastic failure algorithm and the crack softening algorithm in the numerical simulation, and the obtained simulation results coincided well with the experimental results. To this end, the stochastic failure algorithm and crack softening algorithm were also introduced in the numerical simulation process to ensure the reliability of the simulation results.

### 3.2. Simulation Condition

The penetration ability and fragmentation effect of PELE projectile are influenced by the impact velocity of projectile, the inner core and outer casing material, and the target plate material and thickness. In order to contrast and analyze, this paper refers to some parameters in the reference [5]. Specifically, the target plate material is steel, the target plate thickness is 3 mm, the impact velocity of projectile is 938 m/s and 1265 m/s, the inner core material combination types are zero gradient—I, zero gradient—II, positive gradient, negative gradient. The simulation conditions corresponding to the zero gradient—I and zero gradient—II type are mainly used to compare with the other two gradient inner core combination types. Based on the corresponding indicators of the above factors, the simulation condition was designed, as shown, in Table 1.

### 3.3. Material Model and Parameters

AUTODYN software has a huge material database, which is convenient for users to directly retrieve the material model and parameters, and allows users to modify and customize the material model. The selection of material models and constitutive equations in this paper refers to the literature [23,24].

The equation of state for all materials adopts Mie–Grüneisen, which is denoted as the “Shock” equation of state in AUTODYN [25]. In this paper, the material model, including strength and failure, is referenced in the literature [21,22]. All material strength models were chosen as “von Mises” strength model, which only requires a given material shear modulus *G* and flow stress *Y*. The tungsten casing material adopts the principal stress/strain failure model, which considers that the material will fail when the tensile principal strain reaches *ε*_T_, or the tensile principal stress reaches *σ*_T_. The aluminum (Al-6061) core material adopts the principal stress failure model, the polyethylene (PE) core material is not added with failure, and the target plate material (Steel-4340) adopts the plastic strain failure model. In order to ensure that the numerical simulation can truly reflect the failure of materials, it is necessary to add erosion to the simulation. The target plate materials adopt the failure erosion algorithm (Failure), and all other materials adopt the geometric strain erosion algorithm (Geometric Strain). Once the grid reaches the corresponding erosion failure conditions, it will be deleted immediately. All material models and related material parameters are listed in Table 2.

## 4. Analysis and Discussion of the Simulation Results

The damage characteristics of PELE projectile are mainly reflected in its penetration ability and fragmentation effect after penetration. In this paper, the penetration ability of PELE projectile is characterized by the axial residual velocity of projectile, which can reflect the energy loss of the projectile after penetrating the target plate. The larger the value of this parameter, the more favorable the subsequent penetration and damage. In addition, the fragmentation effect of PELE projectile is characterized by the radial scattering velocity of fragments and the residual length of casing. The radial scattering velocity of fragments reflects the damage area of the projectile after penetrating the target. The larger the value, the larger the damage area to the subsequent target. Here, the residual length of casing is defined as the length from the section where no axial crack is generated in the casing to the projectile rear. The smaller the value, the higher the conversion rate of fragments, that is, more casings will be converted into fragments.

In order to obtain the influence law of different inner core combination types on the penetration ability and fragmentation effect, several working conditions were designed to study. In the following analysis and discussion, the corresponding results of zero gradient—I type and zero gradient—II type were mainly used as the basic comparison data to analyze the influence law of the other two gradient inner cores. The results of the projectile crushing of the 8 groups of working conditions at *t* = 100 μs were statistically summarized, and the broken projectile shape under each working condition were obtained as shown in Figure 7. Next, the results of different simulation conditions will be analyzed and discussed in detail, sequentially.

### 4.1. Influence of Inner Core Combination Type on the Penetration Ability of PELE Projectile

For the penetrating projectile, the penetrating ability of projectile is mainly determined by factors such as the projectile velocity, the projectile material, the target plate material, the target plate thickness, and the posture of projectile–target intersection. It can be seen from the previous working conditions that the casing material, the target plate material, the target plate thickness, and the posture of projectile–target intersection have been determined. Therefore, we only need to study the effect of different inner core combination types on the penetration ability of projectile under different initial impact velocities in this section. As mentioned above, the penetration ability of projectile mainly depends on the axial residual velocity of projectile. By analyzing the simulation results, the velocity change curves of projectile during the penetration process were obtained, as shown in Figure 8. In order to more intuitively show the velocity state change during the penetration process, taking the #3 working condition as an example, the velocity distribution state of the #3 working condition at several typical moments is shown in Figure 9.

The velocity change curves of the entire projectile during the penetration process can be seen, very intuitively, from Figure 8. In addition, the velocity distribution state of PELE projectile at various moments can be clearly seen from the numerical simulation as shown in Figure 9. Therefore, the velocity change of projectile can be reasonably explained by combining Figure 8 and Figure 9. At the moment when the projectile impacts the target plate, the velocity of projectile decreases rapidly, due to the impedance of the target plate. At the same time, shock waves will be generated in the projectile and target plate, respectively. After the shock wave in the projectile is subjected to back-and-forth reflection, the velocity of projectile will be raised, to some extent. Subsequently, when the projectile perforates the target plate, the velocity difference between the target plate plug and the projectile will cause the target plate plug to squeeze the inner core, which, in turn, reduces the velocity of the entire projectile. Finally, when the extrusion of the target plate plug and the inner core is completed, the velocity of entire projectile and target plate plug are consistent. Since the influence of air resistance is not considered in the whole simulation process, that is, the attenuation effect of the projectile in the air is neglected, the final axial residual velocity of the projectile tends to a stable value.

The commonalities of the penetration velocity changes of the four different inner core combinations are given above, and the differences between them will be discussed from the perspective of velocity loss. The velocity loss of projectile can also be obtained from the analysis of Figure 8, as shown in Table 3.

Based on the previous analysis, it can be seen that the greater the velocity loss means that the smaller the surviving velocity of projectile after perforating the target plate, the more unfavorable the subsequent target damage. Therefore, it is desirable that the velocity loss of projectile after perforating the target plate is as small as possible. As can be seen from Table 3, when the impact velocity is 938 m/s, the order of velocity loss is #7 ≈ #1 < #5 < #3 (Negative gradient ≈ Zero gradient—I < Positive gradient < Zero gradient—II); when the impact velocity is 1265 m/s, the order of velocity loss is: #8 < #2 < #6 < #4 (Negative gradient < Zero gradient—I < Positive gradient < Zero gradient—II). Based on the simulation results of two different initial impact velocities, it can be approximately considered that the PELE projectile with negative gradient inner core combination has the least velocity loss after perforating the target plate, which is conducive to the subsequent target damage.

### 4.2. Influence of Inner Core Combination Type on the Fragmentation Effect of PELE Projectile

The fragmentation effect of PELE projectile is mainly reflected in the radial scattering velocity of fragments and the effective fragment mass. The radial scattering velocity of fragments or the radial scattering radius of fragments determines the damage area of fragments. The effective fragment mass is characterized by the residual length of projectile. The shorter the residual length of projectile, the higher the fragment conversion rate, that is, more casings will be converted to fragments; otherwise, the lower the fragment conversion rate. By extracting the result data, the maximum radial scattering velocity of fragment corresponding to different working conditions and the radial scattering radius of fragment at *t* = 100 μs can be rapidly obtained, as shown in Figure 10 and Figure 11, respectively.

As shown in Figure 10 and Figure 11, both the radial scattering velocity and radial scattering radius of fragments increase significantly with the increase of initial impact velocity. The reason for this phenomenon is that the increase of initial impact velocity means that more axial velocity is converted into radial velocity in the penetration process. The increase of radial scattering velocity of fragments will inevitably lead to the increase of radial scattering radius of fragments. For the single inner core material, when the inner core material is Al, the radial scattering velocity and radial scattering radius of fragments are significantly better than those of the PE core material. Due to the strength of Al is greater than the strength of PE, when the PE inner core is subjected to impact, more energy will be consumed by its own deformation. While the compressibility of Al inner core is relatively weak, the axial impact compression potential energy can be more converted into the radial force on the outer casing. In addition, the two parameters corresponding to the negative gradient inner core combination type are optimal among the four types of inner core combinations. After careful analysis, it is also found that the radial scattering velocity and radius of fragments of different inner core types depend mainly on the front inner core material. In other words, when the initial impact velocity is constant, the two parameters of negative gradient inner core approximate to the zero gradient—II inner core, and that of the positive gradient inner core approximates to the zero gradient—I inner core.

For the fragmentation effect of PELE projectile after perforating the target, it is necessary to consider both the effective fragment mass and the radial velocity of fragments. That is to say, if only the fragments have a high radial velocity and the effective mass is low, it is not the best. Therefore, this paper hopes to find the best inner core combination type by characterizing the effective fragment mass with the residual length of projectile, which can take into account the two indicators. Based on the numerical simulation results, the residual length of projectile, the radial velocity and radial scattering radius of fragments at *t* = 100 μs can be obtained, as shown in Table 4.

From the point of view of the terminal damage effect, the radial velocity of fragments should be as large as possible (the radial scattering radius of fragments should be as large as possible), and the residual length of projectile should be as small as possible (the effective fragments mass should be as large as possible). Based on these principles, it can be quickly determined, from Table 4, that the negative gradient inner core combination type is the best of the four types. In order to explain the action mechanism of the four kinds of inner core combination more intuitively and clearly, we will draw support from the numerical simulation to analyze the process of projectile–target interaction. Taking four groups of working conditions with impact velocity of 1265 m/s as an example, the projectile–target interaction states under four groups of working conditions at different typical moments are obtained, as shown in Figure 12.

As can be seen from Figure 12, the shape of the target plug formed by the projectile perforating the metal target plate is related to the inner core material, and it can be simply regarded as that the shape of the plug mainly depends on the front inner core material. For the positive gradient type and the zero gradient—I type, the longitudinal section of the target plug is similar to a sector, and the lateral width is larger. For the negative gradient type and the zero gradient—II type, the longitudinal section of the target plug is similar to a trapezoid, and the lateral width is smaller. The different shapes of target plug lead to great differences in the interaction between the plug and inner core. Since the target plug of the zero gradient—II combination type has a wide cross-section and the inner core material is relatively hard, it inevitably results in a larger radial velocity and radial scattering radius of fragments. In addition, because the target plug cannot continue to interact with the subsequent inner core, the casing near the projectile rear cannot be converted into fragments, which will inevitably make the effective mass of fragments lower. On the contrary, for the zero gradient—I combination type, the cross-section of target plug is narrow, and the inner core material is relatively soft, and the plug can interact with the inner core until the projectile rear, so the radial velocity and radial scattering radius of fragments are relatively small, but the effective mass of fragments is higher.

Therefore, it can be seen, intuitively, that the fragmentation effects of two PELE projectiles with single inner core are not ideal, because the radial velocity of fragments (radial scattering radius of fragments) and the effective mass of fragments cannot be taken into account simultaneously. To this end, two kinds of gradient inner core combination type, composed of Al and PE, were designed to find the projectile structure which can consider both the radial velocity of fragments and the effective mass of fragments. The corresponding fragment effects of the two gradient inner core projectiles are shown in Figure 12c,d. As can be seen from Figure 12c,d, the action process between the target plug and inner core of the positive gradient inner core combination type projectile is that the plug compresses the front inner core PE first, and then compresses the rear inner core Al after the compression process of PE is finished. Meanwhile, the action process between the target plug and inner core of negative gradient inner core combination-type projectile is that the target plug compresses the front inner core Al and rear inner core PE at the same time. The above phenomenon indicates that there are differences between the positive gradient and negative gradient inner core in the inner core compressor mechanism, and the fragmentation effect of negative gradient inner core combination type is better than that of positive gradient inner core combination type.

More importantly, after comparing the fragmentation states of the four different inner core combination types at *t* = 100 μs, it can be found that the negative gradient inner core combination type is the best of the four combination types, from the radial scattering radius of fragments and the effective mass of fragments. Obviously, the parameters used to characterize the penetrating ability and fragmentation effect of PELE projectiles have been obtained. Therefore, in order to better support the conclusions obtained in this paper, the simulation results of the single core PELE projectile in this paper were compared with the Paulus’ test results. Based on the results obtained in the above two sections, the error analysis of numerical simulation in this paper and the experimental results in the reference [5] can be obtained, as shown in Table 5.

According to Table 5, the error between the numerical simulation results and the experimental results in reference [5] is less than 3.5% for the axial residual velocity of projectile. It indicates that the simulation results on the penetration ability of PELE projectile are in good agreement with the experimental results. For the radial scattering velocity of fragments, the error of the two working conditions (PE-1265, Al-938) is within 1.5%. The error of the working condition (PE-938) is 16.0%, but the error between the test result and the theoretical calculation result corresponding to this working condition in the literature [5] is as high as 39.4%. In addition, the error of the working condition (Al-1265) is 18.5%, but the radial scattering velocity obtained in this paper is consistent with the theoretical settlement speed in reference [5], both of which are 288 m/s. Therefore, it can be seen that the measured radial scattering velocity of fragments in reference [5] has a certain fluctuation. It can be known, from the analysis, that it is relatively easy to accurately measure the axial residual velocity of the projectile in the experiment, but it is difficult to capture the fragment with the largest radial scattering velocity when measuring the radial scattering velocity of fragments, which results in the large fluctuation of the measured radial scattering velocity of fragments. In addition, considering that there is a certain deviation between the impact velocity of projectile in this paper and the actual velocity of the projectile in the experiment, it can be considered that the error analysis of the numerical simulation results in this paper is acceptable, and the data obtained can well support the conclusions.

In order to explain, more clearly, why the comprehensive performance of negative gradient inner core projectile is superior to other inner core combination projectiles, some reasonable explanations will be given from the physical mechanism. For a single inner core projectile, there is no reflection and transmission of stress wave in the homogeneous material. During the process of penetrating target with multi-core material PELE projectile, when the stress wave propagates to the interface of different core materials, reflection and transmission will occur. When the inner core material adopts the negative gradient combination type, the transmitted wave and the reflected wave stress amplitude are lower than the incident wave stress amplitude.

In terms of penetrating ability, when the front inner core is Al, because its strength is higher than PE and the Poisson’s ratio is smaller than PE, the energy dissipation is less when it impacts the target plate, so the negative gradient core combination is better than zero gradient—I and positive gradient. For the negative gradient and zero gradient—II, according to Figure 12, the negative gradient inner core projectile is compressed both Al core and PE core, while the zero gradient—II inner core projectile is compressed from the front stage and propagates gradually to the projectile rear. It makes the cross-section of the impact end of negative gradient inner core projectile relatively small when penetrating the target plate. Therefore, the resistance of the projectile is also small, which ultimately leads to the negative gradient inner core projectile having a better penetration ability.

In terms of the fragmentation effect, when the front inner core is Al, since the compressibility of Al is less than that of PE, when the projectile impacts the target plate, more energy is converted into the radial force of the casing. Therefore, the radial scattering velocity and radius of fragments corresponding to the negative gradient and zero gradient—II are higher than that of the positive gradient and zero gradient—I after the projectile perforates the target plate. For the negative gradient and zero gradient—II, since the strength of PE is lower than Al, the interaction distance between the target plug and the negative gradient core is much longer than that of zero gradient—II, which leads to the residual length of negative gradient inner core projectile being necessarily lower than that of zero gradient—II. In other words, the fragment conversion rate of negative gradient inner core projectile is higher than that of zero gradient—II. Therefore, it can be considered that the fragmentation effect of the negative gradient inner core projectile is the best.

### 4.3. Penetration Ability and Fragmentation Effect of Different Inner Core Combinations PELE Projectile under the Same Initial Kinetic Energy

The simulation conditions of the above two sections are based on the same impact velocity. However, in practical engineering applications, the projectiles, after launch, usually have the same initial kinetic energy. Although there are differences in the inner core densities of the four combinations in this paper, the difference is relatively small. In order to facilitate the comparative analysis of the radial scattering velocity of fragments after penetration, the above two sections are all conducted under the same initial impact velocity. However, in order to make the conclusions more scientific and rigorous, this paper designs several additional working conditions with the same initial kinetic energy, as shown in Table 6. Several other working conditions were designed based on the #2 working condition (PE inner core, impact velocity 1265 m/s, zero gradient—I). According to the known conditions, the kinetic energy of the projectile corresponding to the 2# working condition can be calculated to be 7.83 × 10^4^ J. Thus, with the same initial kinetic energy, the initial impact velocities of # 9, # 10, and # 11, can be calculated as 1237, 1251, and 1251 m/s, respectively.

According to Section 4.2, the fragmentation effect of PELE projectile is mainly characterized by the radial scattering velocity of fragments, the radial scattering radius of fragments, and the residual length of projectile. However, under the same initial kinetic energy, the initial impact velocity of projectile is not the same, so it is not appropriate to continue to take the radial scattering velocity of fragments as a characterization parameter. Therefore, in this section, the radial scattering radius of fragments and the residual length of projectile are taken as the characterization parameters of fragmentation effect, and the penetration ability of projectile is still characterized by the velocity loss of projectile. By analyzing the simulation results of the four groups of working conditions, the front view and top view of projectile, corresponding to the four groups of working conditions at *t* = 100 μs, can be obtained as shown in Figure 13. In addition, the velocity loss of projectile, radial scattering radius of fragments, and residual length of projectile under four working conditions at *t* = 100 μs, are obtained by statistics as shown in Table 7.

According to Figure 13 and Table 7, under the same initial kinetic energy, the velocity loss and residual length of projectile corresponding to #11 condition (negative gradient) are still the smallest, and the radial scattering radius of fragments corresponding to #11 condition (negative gradient) is still the largest among the four inner core combination types. Therefore, this result also indicates that the penetration ability and fragmentation effect of PELE projectile with negative gradient inner core are better than those of other three inner core combination types, which further validates the correctness of the above conclusions.

## 5. Conclusions

As a new type of penetrating projectile, the PELE projectile mainly considers the penetration ability and fragmentation effect of projectile when it damages the target. By analyzing the casing fracture mechanism, it is found that the inner core material type will significantly affect the damage characteristics of projectile. To this end, four different inner core combination types were designed in this paper, namely, zero gradient—I type (PE), zero gradient—II type (Al), positive gradient type (PE + Al), and negative gradient type (Al + PE). The research was carried out by means of a numerical simulation method, which is relatively mature and can reflect the actual crushing process of PELE projectile. By analyzing the simulation results, the following conclusions can be drawn:(1)The axial residual velocity of projectile can reflect the damage capability of the projectile to the subsequent target, so it is desirable that the velocity loss of the projectile during the target penetration is as small as possible. As shown in Table 3, when the impact velocity is 938 m/s, the velocity loss corresponding to the negative gradient inner core combination is 2.1%; when the impact velocity is 1265 m/s, the velocity loss corresponding to the negative gradient inner core combination is 1.3%; they are all the minimum values under their respective impact conditions. In other words, the axial residual velocity corresponding to the negative gradient inner core combination is the largest, which means that it is superior to the other three types of projectiles in penetration ability.(2)The maximum radial velocity of fragments and the maximum radial scattering radius of fragments directly reflect the damage area of fragments, so it is hoped that the two parameters are as large as possible. As shown in Table 4, when the impact velocity is 938 m/s, the two parameters corresponding to the negative gradient inner core combination are 190 m/s, 21.9 mm; when the impact velocity is 1265 m/s, the two parameters corresponding to the negative gradient inner core combination are 296 m/s, 33.1 mm. Obviously, the two parameters corresponding to the negative gradient inner core combination are higher than the other three types of projectiles under different impact conditions. With the increase of impact velocity, the two parameters corresponding to the four types of projectiles are significantly enhanced, which means that the coverage area of fragments is increased, and the lateral effect is improved. However, at the same impact velocity, these two parameters mainly depend on the front inner core material, that is, if the front inner core is the same material, the values of the two parameters are not much different.(3)The fragmentation effect of PELE projectile should be concerned not only with the maximum radial velocity of fragments and the maximum radial scattering radius of fragments but, also, with the effective mass of fragments. In this paper, the residual length of projectile is used to characterize the effective mass of fragments, that is, the shorter the residual length of projectile means that the more casings will be converted into fragments. As shown in Table 4, when the impact velocity is 938 m/s, the parameter corresponding to the negative gradient inner core combination is 13.5 mm; when the impact velocity is 1265 m/s, the parameter corresponding to the negative gradient inner core combination is 5.8 mm. The residual length of projectile corresponding to negative gradient inner core combination is the shortest of the four types, that is to say, the fragment conversion rate corresponding to negative gradient inner core combination type is the highest.(4)The different types of inner core materials will lead to the different shape of the target plug, which will, in turn, lead to the different interaction process between the plug and the inner core. In addition, it is also found that the radial velocity and radial scattering radius of fragments can be improved by properly increasing the strength of the front inner core material, and the residual length of projectile can be reduced, or the fragment conversion rate can be enhanced by properly reducing the strength of the rear inner core material.

In summary, the PELE projectile corresponding to the negative gradient inner core combination type is the best, in terms of the penetration ability and fragmentation effect, which can better meet the requirements of tactical indicators. In the follow-up research, the structure of negative gradient inner core combination type of projectile will be further optimized to provide a reference for engineering applications.

## Figures and Tables

**Figure 1 materials-11-02389-f001:**
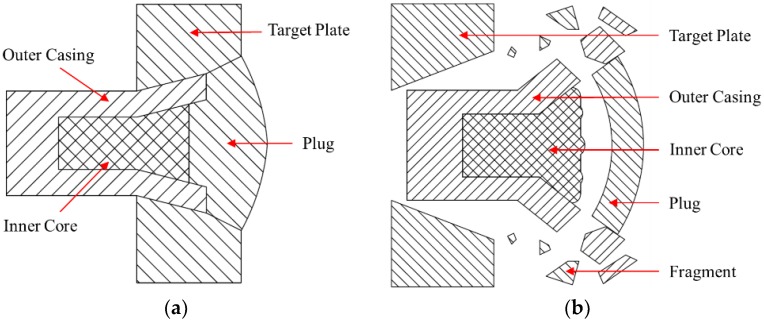
Different penetration state of PELE (penetration with enhanced lateral efficiency) projectile: (**a**) Initial penetration state; (**b**) Completely perforating state.

**Figure 2 materials-11-02389-f002:**
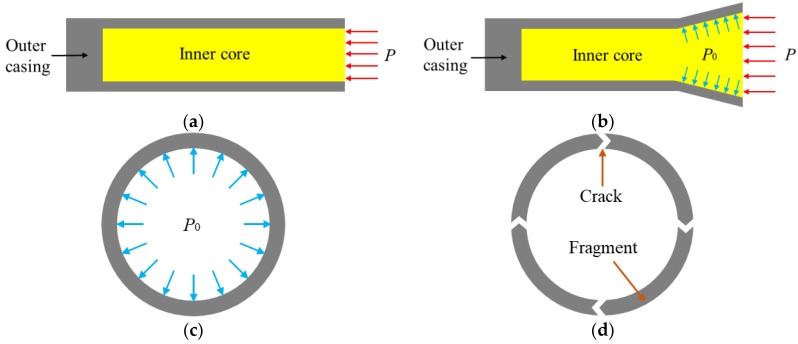
Simplified force state of the projectile and the equivalent ring at different moments: (**a**) Force diagram of the projectile impacting the target at the initial moment; (**b**) Force diagram of the inner core when it is radially expanded; (**c**) Force diagram of the equivalent ring; (**d**) Simplified schematic diagram of the outer casing fracture.

**Figure 3 materials-11-02389-f003:**
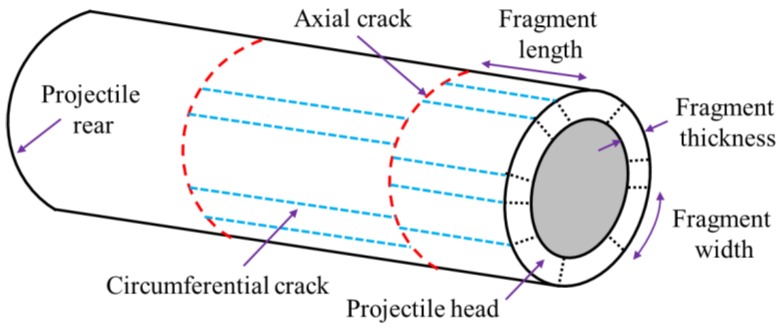
Definition of each parts of PELE projectile and fragments.

**Figure 4 materials-11-02389-f004:**
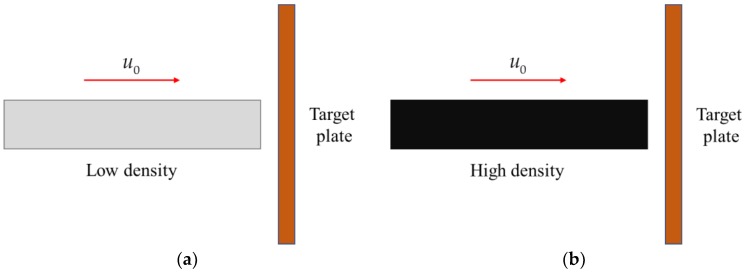
Different combination types of inner core: (**a**) Zero gradient—I; (**b**) Zero gradient—II; (**c**) Positive gradient; (**d**) Negative gradient.

**Figure 5 materials-11-02389-f005:**
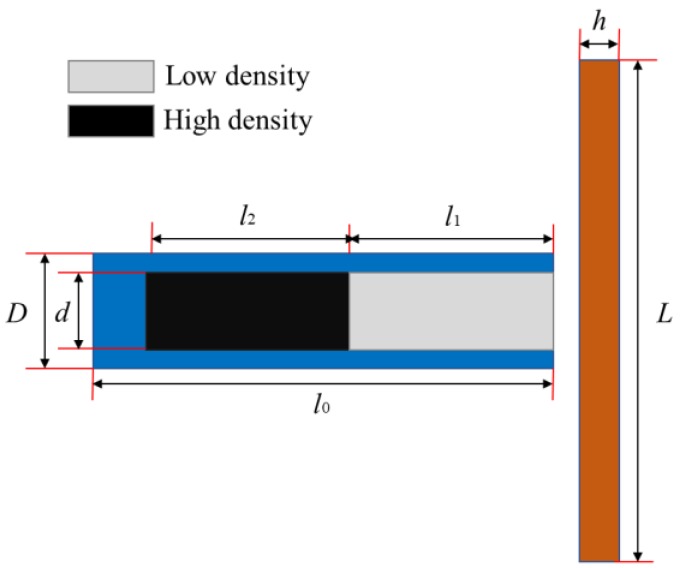
Geometric schematic diagram of the entire model.

**Figure 6 materials-11-02389-f006:**
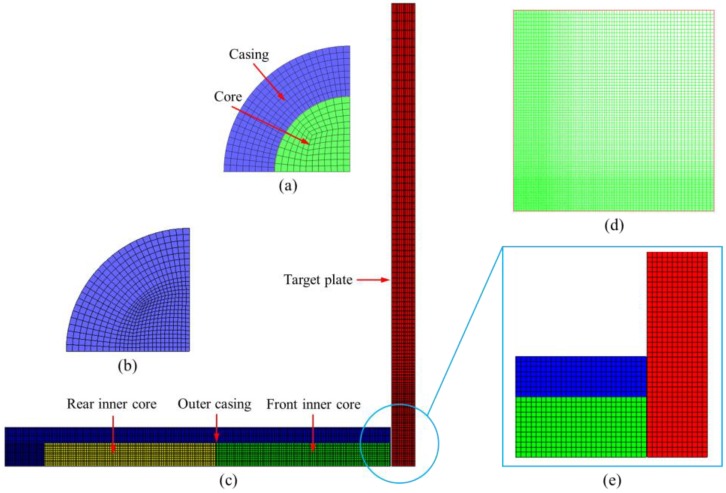
Schematic diagram of the finite element model. (**a**) Projectile head, (**b**) projectile rear, (**c**) 1/4 structural model, (**d**) target plate grid, and (**e**) impact region.

**Figure 7 materials-11-02389-f007:**
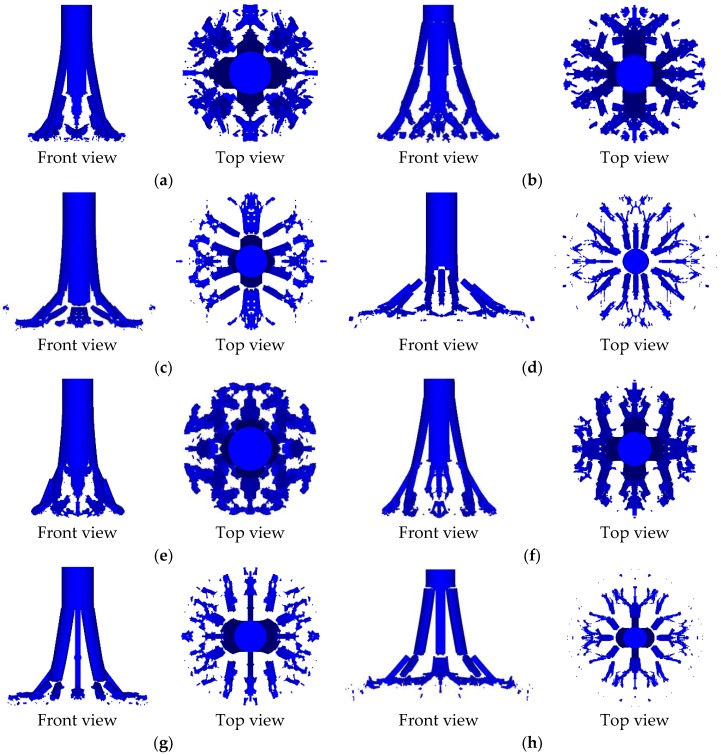
Front view and top view of the projectile of each working condition at *t* = 100 μs. (**a**) #1 projectile, (**b**) #2 projectile, (**c**) #3 projectile, (**d**) #4 projectile, (**e**) #5 projectile, (**f**) #6 projectile, (**g**) #7 projectile, and (**h**) #8 projectile.

**Figure 8 materials-11-02389-f008:**
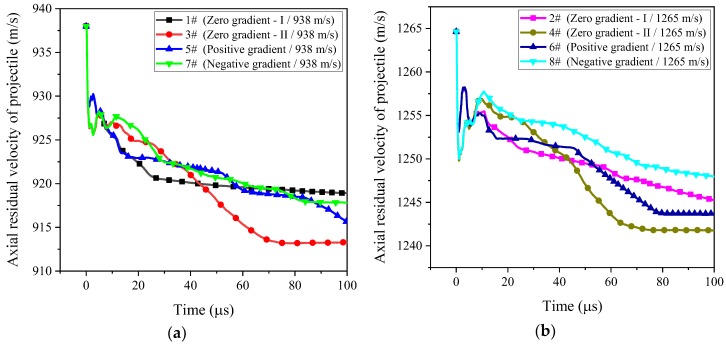
Axial residual velocity of projectile corresponding to four inner core combination types under different impact velocities. (**a**) *u*_0_ = 938 m/s, (**b**) *u*_0_ = 1265 m/s.

**Figure 9 materials-11-02389-f009:**
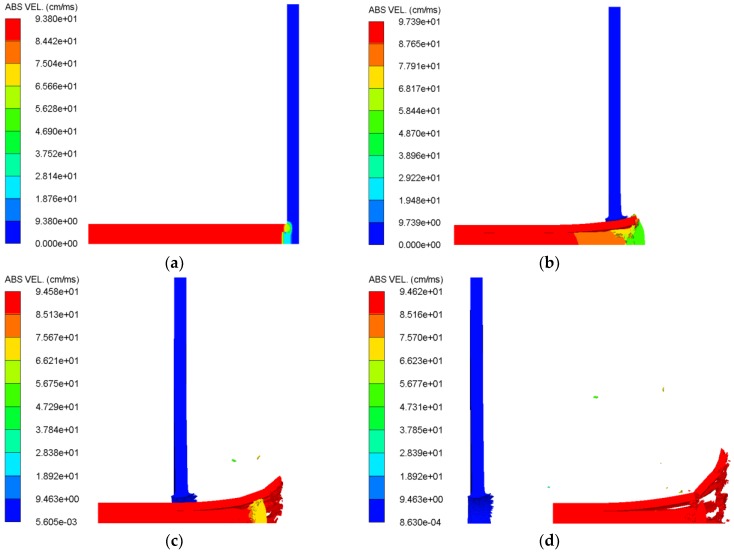
Velocity distribution of projectile and target plate at typical time corresponding to #3 working condition. (**a**) *t* = 1 μs, (**b**) *t* = 12 μs, (**c**) *t* = 34 μs, and (**d**) *t* = 76 μs.

**Figure 10 materials-11-02389-f010:**
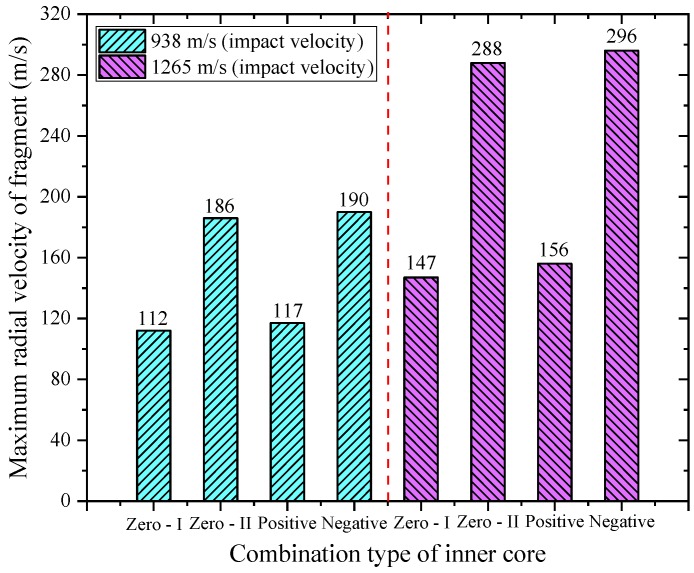
Maximum radial velocity of fragment corresponding to different working conditions.

**Figure 11 materials-11-02389-f011:**
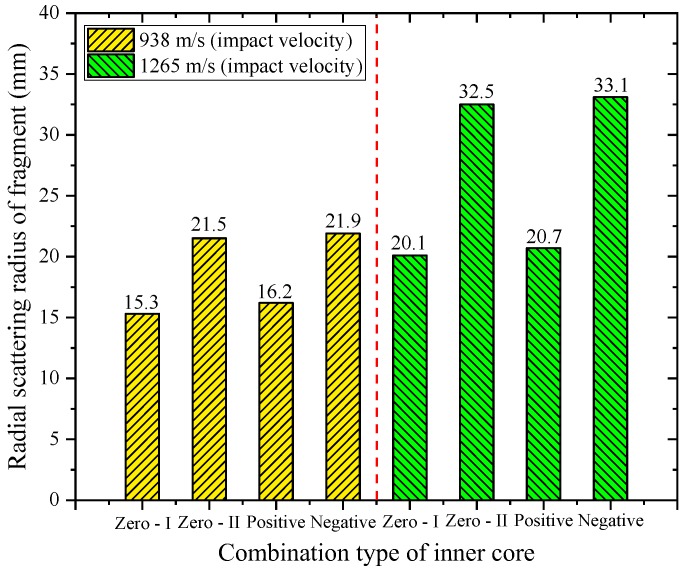
Radial scattering radius of fragment corresponding to different working conditions.

**Figure 12 materials-11-02389-f012:**
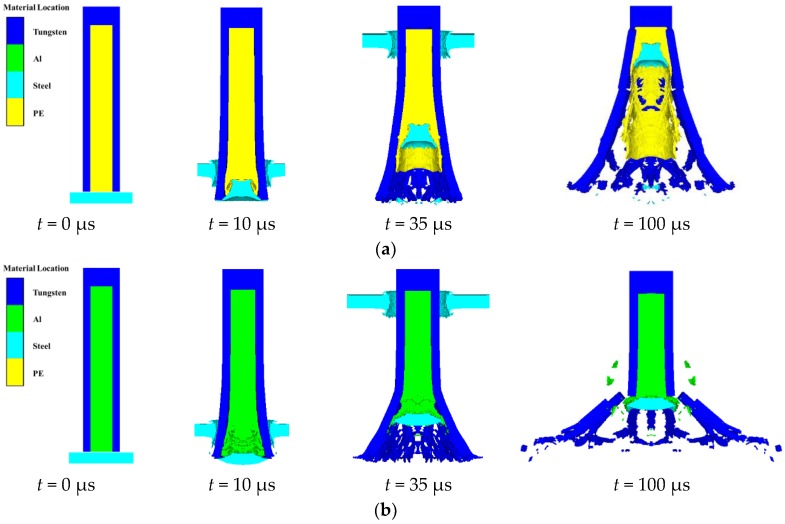
The projectile–target interaction states under four groups of working conditions at different typical moments. (**a**) Zero gradient—I type; (**b**) Zero gradient—II type; (**c**) Positive gradient type; (**d**) Negative gradient type.

**Figure 13 materials-11-02389-f013:**
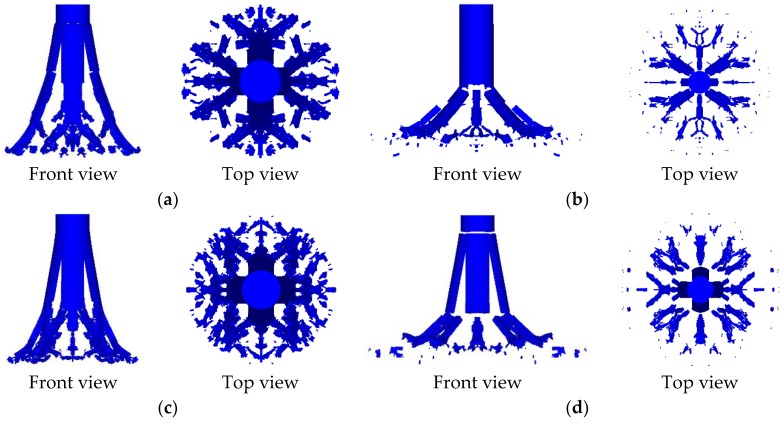
Front view and top view of the projectile at *t* = 100 μs. (**a**) #2 projectile, (**b**) #9 projectile, (**c**) #10 projectile, and (**d**) #11 projectile.

**Table 1 materials-11-02389-t001:** Simulation condition.

Condition Number	Target Material	Target Thickness	Inner Core Combination Type	Impact Velocity	Front Core Material	Rear Core Material
#1	Steel	3 mm	① Zero gradient—I (*l*_1_ = 45 mm, *l*_2_ = 0 mm)	938 m/s	PE	--
#2				1265m/s		
#3			② Zero gradient—II (*l*_1_ = 0 mm, *l*_2_ = 45 mm)	938 m/s	--	Al
#4				1265m/s		
#5			③ Positive gradient (*l*_1_ = *l*_2_ = 22.5 mm)	938 m/s	PE	Al
#6				1265m/s		
#7			④ Negative gradient (*l*_1_ = *l*_2_ = 22.5 mm)	938 m/s	Al	PE
#8				1265 m/s		

**Table 2 materials-11-02389-t002:** Material model and parameters in the numerical simulation.

Variable	Material			
	Tungsten	Al-6061	PE	Steel-4340
EOS	Shock	Shock	Shock	Shock
Strength Model	von Mises	von Mises	von Mises	von Mises
Failure Model	Principal Stress/Strain	Principal Stress	--	Plastic Strain
Erosion	Geometric Strain	Geometric Strain	Geometric Strain	Failure
*ρ*_0_ (g/cm^3^)	18	2.65	0.92	7.823
*c*_0_ (km/s)	4.03	5.24	2.9	4.57
*s*	1.237	1.4	1.48	1.49
Grüneisen Coefficient	--	1.97	1.6	--
*C*_p_ (J/kg·K)	--	885	2300	--
Shear Modulus *G* (GPa)	139.02	27.5	0.13	77
Yield Stress *Y* (GPa)	1.5	0.3	0.02	0.8
Principle Tensile Stress *σ*_T_ (GPa)	2.8	0.5	--	--
Principle Tensile Strain *ε*_T_	0.035	--	--	--
Fracture Energy *G*_f_ (J/m^2^)	45	--	--	--
Stochastic Variance *γ*	36.5	--	--	--
Geometric Strain	0.6	0.8	1.8	Failure

**Table 3 materials-11-02389-t003:** Velocity loss of projectile under different working conditions.

Condition Number	Impact Velocity (m/s)	Inner Core Combination Type	Axial Residual Velocity of Projectile (m/s)	Velocity Loss of Projectile (%)
#1	938	Zero gradient—I	919	2.0
#3		Zero gradient—II	913	2.7
#5		Positive gradient	916	2.3
#7		Negative gradient	918	2.1
#2	1265	Zero gradient—I	1245	1.6
#4		Zero gradient—II	1242	1.8
#6		Positive gradient	1244	1.7
#8		Negative gradient	1248	1.3

**Table 4 materials-11-02389-t004:** Key parameters of the fragmentation effect (*t* = 100 μs).

Condition Number	Impact Velocity (m/s)	Inner Core Combination Type	Radial Velocity of Fragment (m/s)	Radial Scattering Radius of Fragment (mm)	Residual Length of Outer Casing (mm)
#1	938	Zero gradient—I	112	15.3	14.8
#3		Zero gradient—II	186	21.5	32.2
#5		Positive gradient	117	16.2	21.6
#7		Negative gradient	190	21.9	13.5
#2	1265	Zero gradient—I	147	20.1	6.0
#4		Zero gradient—II	288	32.5	26.7
#6		Positive gradient	156	20.7	8.3
#8		Negative gradient	296	33.1	5.8

**Table 5 materials-11-02389-t005:** Working conditions under the same initial kinetic energy.

Inner Core Material	Impact Velocity		Axial Residual Velocity of Projectile (m/s)			Radial Velocity of Fragment (m/s)		
	Reference [5]	This Paper	Reference [5]	This Paper	Error (%)	Reference [5]	This Paper	Error (%)
PE	936	938	889	919	3.37	94	112	16.0
	1262	1265	1206	1245	3.23	145	147	1.38
Al	925	938	895	913	2.01	184	186	1.09
	1261	1265	1231	1241	0.81	243	288	18.5

**Table 6 materials-11-02389-t006:** Working conditions under the same initial kinetic energy.

Condition Number	Target Material	Target Thickness	Impact Velocity	Inner Core Combination Type	Front Core Material	Rear Core Material
#2	Steel	3 mm	1265 m/s	Zero gradient—I	PE	--
#9			1237 m/s	Zero gradient—II	--	Al
#10			1251 m/s	Positive gradient	PE	Al
#11			1251 m/s	Negative gradient	Al	PE

**Table 7 materials-11-02389-t007:** Key parameters of the fragmentation effect under the same initial kinetic energy (*t* = 100 μs).

Condition Number	Impact Velocity (m/s)	Inner Core Combination Type	Velocity Loss of Projectile (%)	Radial Scattering Radius of Fragment (mm)	Residual Length of Outer Casing (mm)
#2	1265	Zero gradient—I	15.8	20.1	6.0
#9	1237	Zero gradient—II	13.1	31.1	24.4
#10	1251	Positive gradient	16.8	19.4	10.6
#11	1251	Negative gradient	12.3	32.3	4.7

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
