# Peer review of "Damage Characteristics of PELE Projectile with Gradient Density Inner Core Material"

_materials, 2018, doi:10.3390/ma11122389_

Reviewer 1 Report

The manuscript is interesting and shows results worth to publish. 

I recommend to authors remove the word "killing"  from the manuscript and smoothen this issue somehow.

Author Response

Dear reviewer,

Thank you very much for your approval of the manuscript. According to the comments, I have revised the manuscript one by one, and all changes have been highlighted by using the “Track Changes” function in Microsoft Word. The detailed responses can be found in the attachment, and I hope that the revised manuscript will satisfy you.

Thanks again for your attention to our manuscript.

Sincerely yours

Liangliang Ding

Reviewer 2 Report

This conclusion: In summary, the PELE projectile corresponding to the negative gradient inner core combination type is the best in terms of the penetration ability and fragmentation effect, which can better meet the requirements of tactical indicators” is without any parameters and should be confirmed with the data from Table 3 and 4.

Author Response

(The authors gave the same response as above.)

Reviewer 3 Report

This paper describes a study using numeric 2D simulations to study the effect of filling material on the lethality of a penetrator with enhanced lateral efficiency (PELE) projectile. The researchers perform a set of 8 numeric simulations examining 4 types of projectiles at 2 impact velocities. From the results of the simulations the authors conclude that a projectile with two material filling – PE at the front followed by Al behind is the most effective combination out of the 4 examined.

As this is purely numerical study one immediately raises the question on the validity of the simulations. Numeric simulations are as good as the material models used, and in this case the extensive deformation and fracture of the projectile (and target) require a rigorous validation. The simulations of the single core material reported here are in fact the reproduction of the Paulus and Schirm experiments (IJIE 2006) and thus a validation of the simulations should be relatively easy. This follows to my next question on the validity of the material models used, the authors did not provide relevant references to the material models (both strength and failure). Will a variation of the numerical material parameters change the results?

The authors found that the projectile with dual core with Al at the front and PE at the back is more effective than other configurations. Comparing between the negative- to positive-gradient projectiles is straightforward. But the single core projectiles have different initial kinetic energy. As this paper aims to address real-life applications, it would be interesting to investigate if the dual-core projectile is more effective than a PE core projectile with the same kinetic energy (not initial velocity).

In general, the paper is clearly written and scientifically sound apart from the issues raised above. This reviewer does feel this work is lacking further physical interpretation and it only reports an engineering observation based on now readily available numeric simulations. While the applications described in this paper are relevant, a deeper understanding between material properties and the application is in place. What is it in the Al that makes its better in the front? These issues need to be better addressed before publication in Materials.

Few minor notes:

The authors used the phrase: “axial kinetic energy into radial kinetic energy” while the meaning of the sentence is clear, kinetic energy is a scalar quantity and does not have a direction.

The authors state that “there are few published papers on the PELE penetration behavior of multicore material”. Please cite these works.

Autodyn is no longer a Century Dynamics software as stated but a part of Ansys Inc.

The following sentence is not clear: “a series of Gaussian points were set in the simulation model”.

Author Response

Dear reviewer,

Thank you very much for your approval of the manuscript. According to the comments, I have revised the manuscript one by one, and all changes have been highlighted by using the “Track Changes” function in Microsoft Word. The detailed responses can be found in the attachment, and I hope that the revised manuscript will satisfy you.

Thanks again for your attention to our manuscript.

Sincerely yours

Liangliang Ding

Round  2

Reviewer 3 Report

Following my previous first comment. The authors did not show in the manuscript that their simulations reproduce Paulus and Schirm experiments. This validation is crucial to support the simulation results and the conclusions as a consequence.

The rest of my comment where adequately addressed. Ones the validation mentioned above added the manuscript can be approved for publishing.

Author Response

Dear reviewer,

Thank you very much for your approval of most of the changes in the revised manuscript. I indeed missed the modification to the question you mentioned (The authors did not show in the manuscript that their simulations reproduce Paulus and Schirm experiments.). Therefore, I made a corresponding supplement in the new revised manuscript (Line 365-389), and all changes have been highlighted by using the “Track Changes” function in Microsoft Word. I hope that the new revised manuscript will satisfy you.

Thanks again for your attention to our manuscript.

Sincerely yours

Liangliang Ding